# FINDING PRIVATE BUGS: DEBUGGING IMPLEMENTATIONS OF DIFFERENTIALLY PRIVATE STOCHASTIC GRADIENT DESCENT

## ABSTRACT

It is important to learn with privacy-preserving algorithms when training data contains sensitive information. Differential privacy (DP) proposes to bound the worst-case privacy leakage of a training algorithm. However, the analytic nature of these algorithmic guarantees makes it difficult to verify that an implementation of a differentially private learner is correct. Research in the field focuses on empirically approximating the analytic bound, which only assesses whether an implementation provides the guarantee claimed for a particular dataset or not. It is also typically costly. In this paper, we take a first step towards providing a simple and lightweight methodology for practitioners to identify common implementation mistakes without imposing any changes to their scripts. Our approach stems from measuring distances between models outputted by the training algorithm. We demonstrate that our method successfully identifies specific mistakes made in the implementation of DP-SGD, the de facto algorithm for differentially private deep learning. These mistakes are improper gradient computations or noise miscalibration. Both approaches invalidate assumptions that are essential to obtaining a rigorous privacy guarantee.

## 1 INTRODUCTION

Machine learning (ML) models trained without taking privacy into consideration may inadvertently expose sensitive information contained in their training data (Shokri et al., 2017; Rahman et al., 2018; Song & Shmatikov, 2019; Fredrikson et al., 2015). Training with differential privacy (DP) (Dwork et al., 2014) emerged as an established practice to bound and decrease such possible leakage. Because differential privacy guarantees are algorithmic, they require modifications to the training algorithm to obtain such a bound. This bound is also known as the privacy budget $\varepsilon$ of the algorithm. Making the necessary modifications can be challenging because practitioners often do not have the DP expertise required to ensure that the implementation is sound and correct, and wrong implementations usually do not "fail loudly" (i.e., they do not block training, nor lead to obvious differences in terms of the performance of the trained models).

In this paper, we approach this problem through testing practices. We focus on the canonical DP learning algorithm, which is the differentially private stochastic gradient descent (DP-SGD) Chaudhuri et al. (2011); Abadi et al. (2016). Established research in the field has considered testing this algorithm but only from an auditing perspective with an external party, e.g., a regulator. Their approach is to interact with the implementation of DP-SGD in a black-box fashion to empirically verify the privacy budget achieved by the algorithm, $\varepsilon$, is the one claimed by its developer (Jagielski et al., 2020; Nasr et al., 2021; Tramer et al., 2022). It is important to note that any discrepancy does not get attributed to the specific mistake(s) made in the implementation. Instead, it simply informs us if an implementation is correct or not.

As we introduce our framework for testing implementations of DP-SGD to identify common failures, we adopt the perspective of the developer themselves. This is orthogonal and, in fact, complementary to prior work on auditing the privacy budget of DP-SGD implementations. Once prior work has identified an incorrect implementation, our framework can be used to help identify the source of the discrepancy. We see two key use cases where this would be beneficial for developers: (1) when

they integrate an existing implementation of DP-SGD in their ML pipeline; and (2) when they write their own implementation of DP-SGD from scratch. While the usefulness of our method is apparent in the latter application scenario, even using an off-the-shelf DP-SGD implementation is not as trivial as it may initially sound. This is due to the fact that open-source implementations of DP-SGD are still nascent and do not always support all modern architectural features of ML pipelines.

Since the outputs of DP-SGD are updates of model parameters, incorrect implementations of it should manifest themselves through differences in the parameter space of the trained ML models. However, the parameter space is typically high-dimensional and the nature of non-convex optimization makes it difficult to evaluate whether the solution yielded by a particular DP-SGD implementation is correct. Therefore, we propose to identify and approximate such differences by evaluating functions defined in the parameter space of the models, or by comparing the models' losses.

Note that the DP-SGD algorithm makes three key modifications to vanilla SGD to obtain differential privacy, each of which represents a potential failure point in its implementation. First, gradients are computed on a per-example basis. This enables the algorithm to isolate the contribution of each training example. Second, the norm of these per-example gradients are clipped. This bounds the sensitivity of the algorithm to each individual training example. Third, gradients are *noised* before they are applied to update the model. This provides the indistinguishability across updates needed to obtain differential privacy.

We introduce a methodology to identify common implementations mistakes for each of these modifications: (1) We observe correct gradient clipping restricts the impact of some training steps on the model and thus the change in loss values caused by such steps. However, this may not be the case if gradient clipping is performed incorrectly or not performed at all. Therefore, we leverage changes in loss values to test correctness of gradient clipping; and (2) Having the noise improperly calibrated to the sensitivity of the training algorithm often results in insufficient indistinguishability in the parameters to obtain differential privacy. Thus we detect improper calibration of the noise added to gradients by measuring the parameter-space distance between models obtained. Note that both of these tests can be performed on checkpoints outputted by the ML pipeline without modifications on the training scripts, which makes them easy to deploy universally.

We validate our approach on standard image classification tasks (i.e., ResNet-20 and Wide ResNet50-2 architectures trained on CIFAR-10 and ImageNet datasets, respectively) and an NLP task (i.e., a Bert model trained on the Stanford Sentiment Treebank v2). Our proposed method is able to detect the three common implementation mistakes we highlighted earlier, namely when: (1) the gradient is evaluated using an aggregate on the mini-batch of data points before clipping; (2) the updates are computed with no clipping; and (3) the additive noise is not calibrated.

In summary, our tests help identify implementation mistakes that developers may not be aware of, and that would otherwise invalidate the privacy guarantees claimed. Our conceptually simple, computationally efficient, dataset-agnostic and model-agnostic tests detect common mistakes in the implementation of gradient clipping and additive noise:

- We characterize the effect of per-example gradient clipping, mini-batch gradient clipping, and no gradient clipping on gradient updates computed by DP-SGD. Based on our analysis, we design a test that detects incorrect gradient clipping by either varying the mini-batch size or the gradient norm bound which clipping is configured to enforce.

- We theoretically demonstrate that the parameter-space distance between models trained with DP-SGD is a function of the scale of noise added to gradients. From this, we obtain a test that detects incorrect noise calibration by varying the value of the gradient norm bound.

- We demonstrate, through extensive experimentation, that a developer can run our tests on their implementations to detect incorrect gradient clipping and incorrect noise calibration in both image and text domains without changing their scripts.

## 2 PROBLEM DESCRIPTION AND RELATED WORK

Differential privacy (Dwork et al., 2006; 2014) (DP) is the gold standard to reason about the privacy guarantees of learning algorithms. A randomized algorithm $A$ is $(\varepsilon, \delta)$-differentially private if its outputs for any $S \in \text{Range}(A)$ and any two neighboring datasets $D$ and $D'$ that differ only by one

record are statistically indistinguishable:

$$\Pr[A(D) \in S] \leq e^{\varepsilon}\Pr[A(D') \in S] + \delta. \tag{1}$$

Machine learning models can be trained under the framework of DP using output perturbation (Wu et al., 2017; Zhang et al., 2017), objective perturbation (Chaudhuri et al., 2011; Iyengar et al., 2019) or gradient perturbation (Bassily et al., 2014; Abadi et al., 2016) methods. In this work, we focus on the gradient perturbation approach, specifically differentially private stochastic gradient, DP-SGD (Abadi et al., 2016). It is known as the de facto differential private learning algorithm (for deep learning). In order to provide DP guarantees, DP-SGD imposes three modifications to vanilla SGD (highlighted in red in Algorithm 1):

- **Per-example gradient computation.** DP-SGD computes the gradient of the loss function with respect to model parameters for each individual example separately to isolate the influence of each training data point on the training algorithm's output (line 5 in Algorithm 1). Note that there is no a priori bound on per-example gradients.
- **Per-example gradient clipping.** DP-SGD thus clips the per-example gradient to a fixed norm $C$, to bound the sensitivity of the gradients to each individual training data point (line 6 in Algorithm 1).
- **Calibrated noise addition.** DP-SGD adds noise to the aggregated clipped gradients before they are applied to update the model parameters. The noise is scaled by the magnitude of $C$ and a noise multiplier $\sigma$ (line 7 in Algorithm 1).

Implementing incorrectly or omitting the above modifications invalidates the guarantee of differential privacy. For example, developers are used to computing gradients for mini-batches of training examples when implementing vanilla SGD as opposed to per-example gradients, i.e., developers may tend to only obtain the aggregated gradient update. This course of action can cause mistakes in the implementation of per-example clipping by either performing the mini-batch aggregation before gradient clipping or missing the clipping altogether. In addition to this, developers, especially those who do not have DP expertise, may forget to scale the noise by $C$ thus implementing uncalibrated noise, thereby rendering the calculation[1] of $\varepsilon$ independent of $C$.

Current approaches to verifying the correctness of a DP-SGD implementation rely on privacy auditing: an attack is designed to obtain an empirical lower bound on the privacy budget of the implemented algorithm (Nasr et al., 2021; Jagielski et al., 2020). This, however, does not identify the source of the mistakes made in the implementation; it simply demonstrates whether ML models trained with DP meet the privacy guarantee claimed by the framework. Furthermore, they are computationally costly as many models need to be trained as part of the attack to ensure statistical validity of the results (Nasr et al., 2021; Jagielski et al., 2020; Tramer et al., 2022). For instance, Tramer et al. (2022) trained 100,000 models on MNIST. These heavy computations limit their applicability to simple tasks and datasets. Finally, it should be noted that expertise in both ML and DP is required to design the attack and, e.g., create poisoned samples required for privacy audits.

## 3    TESTING IMPLEMENTATIONS OF DP-SGD

We address the issues identified in Section 2 by introducing conceptually simple and computationally efficient tests enabling developers to detect the source of mistakes in their DP-SGD implementations. We show that these implementation mistakes manifest themselves through differences in the dependency of gradient updates to DP-related hyperparameters. Therefore, we do not impose any changes to training scripts. Instead, our method only requires one run of the training scripts while setting the value of DP-related hyperparameters such as $C$, $B$, and $\delta$ to particular values. In details, we make the three following assumptions:

1. No real data: the developer should not need real data points for running tests, they should also be able to use synthetic data points.
2. No privacy risk: any gradient computation or model trained as part of the proposed tests does not need to be released; put another way the sole purpose of these tests is to assess the

---

[1]The privacy budget is computed as a function of the mini-batch sampling rate, the number of training steps, and the noise multiplier.

correctness of a DP-SGD implementation. This means the approach does not increase the risk of leakage of private information when these tests were executed using sensitive inputs from real data.

3. No changes: tests should not impose any changes to training scripts, and they should be easy and light to run.

---

**Algorithm 1** DP-SGD

---

**Require:** Dataset $D$, Mini-Batch Size $B$, Gradient Norm Bound $C$, Noise Multiplier $\sigma$, Model Parameters $W$, Loss Function $L$, Learning Rate $\eta$
1: $W_0 \leftarrow$ RandomInitialization()
2: **for** $t \leftarrow 1, \ldots, T$ **do**                                      ▷ Training Steps
3:     $MiniBatch \leftarrow$ RandomlySelectMiniBatches($D, B$)
4:     **for** $b \leftarrow 1, \ldots, B$ **do**             ▷ Iterate over every data point in the mini-batch
5:         $g_b = \nabla_W L(W_{t-1}, MiniBatch[b])$         ▷ Per-example gradient calculation
6:         $\bar{g}_b = g_b / \max(C^{-1}\|g_b\|_2, 1)$         ▷ Per-example gradient clipping
7:     $g = \frac{1}{B}(\sum_b \bar{g}_b + \mathcal{N}(0, (C\sigma)^2))$         ▷ Add calibrated Gaussian Noise
8:     $W_t \leftarrow W_{t-1} - \eta g$
9: **return** $W_T$

---

### 3.1 DETECTING INCORRECT GRADIENT CLIPPING USING LOSS VALUES

Clipping per-example gradients to a fixed norm $C$ is an essential step in DP-SGD to bound the influence of each data point on the final gradients. Two common mistakes in implementing this gradient clipping operation include 1) *no gradient clipping*: forgetting to perform any sort of gradient clipping; and 2) *mini-batch gradient clipping*: aggregating gradients of all data points within each mini-batch first and then clipping the aggregated gradient, instead of clipping the per-example gradients first and then aggregating them. Next, we analyze differences caused by such mistakes and describe how to capture them in a test.

Let $\{(x_i \in \mathbb{R}^N, y_i)\}_{i=1}^B$ be a mini-batch of $B$ data points with per-example gradients $\{g_i\}_{i=1}^B$ for the model $M$. We set the noise multiplier to $\sigma = 0$ (i.e., disabling the addition of noise to the gradients in DP-SGD) to isolate the effect of gradient clipping. The private gradient, i.e., gradient after it has been modified by DP-SGD and aggregated, computed by correct and incorrect implementations of gradient clipping would be:

$$\text{Private Gradient} = \begin{cases} \frac{1}{B}\sum_b \left(g_b / \max(C^{-1}\|g_b\|_2, 1)\right) & \text{Per-example gradient clipping,} \\ \frac{1}{B}\sum g_b & \text{No gradient clipping,} \\ \left(\frac{1}{B}\sum g_b\right) / \max(C^{-1}\|\frac{1}{B}\sum g_b\|_2, 1) & \text{Mini-batch gradient clipping.} \end{cases} \quad (2)$$

Note that we would like our test to be implementable without having to make modifications to the training script, or needing to inspect the (private) gradients directly. Instead of comparing gradients directly, we thus instead leverage the change in loss values:

$$\text{Change in loss values} = Loss(M + \text{optimizer(Private Gradient)}) - Loss(M), \quad (3)$$

which is a function of private gradients.

To capture and highlight differences among these 3 types of gradient clipping in practice, we carefully set the value of hyperparameters, which are only used for computing private gradients but not the rest of Equation 3. This ensures that each hyperparameter yields different changes in loss values.

**Detect no gradient clipping.** Comparing the first line and the second line of Equation 2 demonstrates that the private gradient with no gradient clipping (and thus the changes in loss values) is *always* independent of $C$. Conversely, the private gradient in the per-example gradient clipping is a function of $C$ *when clipping is effective* i.e., $C < \|g_b\|$. Therefore, our test carefully varies the value of $C$ and detects no gradient clipping based on the differences between the changes in loss values across different $C$. As shown in Figure 1 (left plot), the changes in loss values caused by no gradient clipping are invariant to changes in $C$. At the same time, increasing $C$ increases changes in

loss values in the case of per-example gradient clipping until the gradient norm bound $C$ becomes larger than the unclipped gradients (i.e., the clipping operation becomes a "no op").

**Detect mini-batch gradient clipping.** When clipping is effective, the model update in per-example gradient clipping (first line of Equation 2) is different than the model update in mini-batch gradient clipping (second line of Equation 2). This is because clipping occurs before the aggregation in the former, while the clipping occurs after the aggregation in the latter. To ensure that the clipping operation does not become a "no op" for both per-example gradient clipping and mini-batch gradient clipping, our test proposes to set the effective clipping as:

$$C < \min\{\|g_b\|_2\}, \quad \forall g_b \neq 0 \quad \text{and} \quad C < \|\frac{1}{B}\sum g_b\|_2. \tag{4}$$

Combining Equation 2 and Equation 4, we have:

$$\text{Private Gradient} = \begin{cases} \frac{C}{B}\sum_b \frac{g_b}{\|g_b\|_2}, & \forall g_b \neq 0 \quad \text{Per-example gradient clipping,} \\ \frac{C\sum_b g_b}{\|\sum_b g_b\|_2} & \text{Mini-batch gradient clipping.} \end{cases} \tag{5}$$

In order to (1) ensure the intersection of ranges of $C$ in Equation 4 is not empty; (2) have control over zero and non-zero gradients; and (3) to cancel out the effect of other elements, except $B$, on the differences between their model updates, we synthesized a mini-batch of data points in which all data points have zero gradients except one. Our data synthesizer receives a mini-batch size $B$ and a feature dimension $N$ as inputs, and creates $\{(x_i \in \mathbb{R}^N, y_i)\}_{i=1}^B$, where $(x_B, y_B)$ is the data point with non-zero gradient $g_B$ such that $\|g_B\| \gg C$, and $\{(x_i, y_i)\}_{i=1}^{B-1}$ are the data points with zero gradients. It is worth to note here it does not affect our approach as to what the features (i.e., $\{x_i\}_{i=1}^B$) of the inputs are. Instead, the former is achieved by setting $y_B = -\alpha M(x_B)$ where $\alpha \gg 1$ (*e.g.*, $\alpha = 10$) to have a non-zero per-example gradient with a larger norm than $C$, thus ensuring the presence of clipping; and the latter (i.e., data points with zero gradients) is achieved by setting the labels of the other $B - 1$ data points such that $\{y_i = \arg\min_{y_i} L(y_i, M(x_i))\}_{i=1}^{B-1}$ so that the loss values (and subsequently the gradients) of this set of data points with the labels are minimized. Since the loss function is a mapping from output space of the model (which is usually low-dimensional) to a scalar, solving y analytically or numerically should be easy. For commonly used loss functions like cross-entropy loss and mean square error, the solution would be $\{y_i = M(x_i)\}_{i=1}^{B-1}$.

Returning to Equation 5, we now have $g_b = 0$ for all points in $\{x_i\}_{i=1}^{B-1}$. The only term left in each sum over $b$ is the term that corresponds to the $B$-th data point. We thus have (see Appendix A for detailed derivation):

$$\text{Private Gradient} = \begin{cases} \frac{Cg}{B\|g_B\|_2} & \text{Per-example gradient clipping,} \\ \frac{Cg}{\|g_B\|_2} & \text{Mini-batch gradient clipping.} \end{cases} \tag{6}$$

Equation 6 demonstrates that the private gradients (and thus the changes in loss values) in mini-batch gradient clipping is independent of the mini-batch size $B$ while increasing $B$ decreases the magnitudes of private gradients in per-example gradient clipping. By leveraging this observation, our test varies the mini-batch size $B$ of the synthesized mini-batch to detect mini-batch gradient clipping as shown in Figure 1 (right plot). It is expected that the changes in loss values in the case of mini-batch gradient clipping always has the same value irrespective of the mini-batch size $B$. In contrast, magnitudes of the per-example clipped private gradient decreases as $B$ increases since the averaging happens after clipping, so smaller changes in loss values are expected. Note that the synthesized data point with a non-zero per-example gradient $((x_B, y_B))$ must be kept the same across runs with different values of $B$. This way, the values for $C$, $g$ (and $\|g\|_2$) are kept constant across different runs so that only the values for $B$ in Equation 6 would vary as independent variables.

### 3.2 DETECTING INCORRECT NOISE CALIBRATION USING MODEL DISTANCE

Based on the actions we perform in the previous sections, we can detect common mistakes in the clipping implementation. Now, we describe a test to identify mistakes in the calibration of noise added by DP-SGD to the clipped gradient. Recall from Section 2, for noise to be correctly calibrated, the variance of the Gaussian noise added to the clipped gradients needs to depend on the gradient

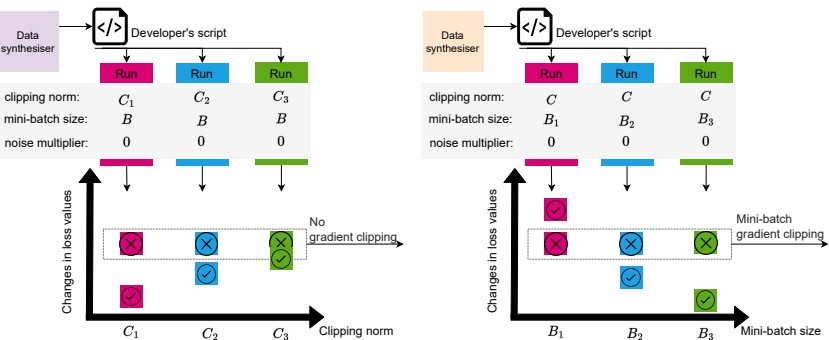

Figure 1: Overview of our test for detecting no gradient clipping (left) and mini-batch gradient clipping (right) in DP-SGD implementations. Observe how gradient updates with per-example gradient clipping overlaps with gradient updates computed without gradient clipping when the gradient norm bound is large.

norm bound $C$. Specifically, we are interested in distinguishing the following private gradients:

$$\text{Private Gradient} = \begin{cases} \frac{1}{B}\left(\sum_b g_b / \max(C^{-1}\|g_b\|_2,\ 1) + \mathcal{N}(0,\ (C\sigma)^2)\right) & \text{Calibrated noise,} \\ \frac{1}{B}\left(\sum_b g_b / \max(C^{-1}\|g_b\|_2,\ 1) + \mathcal{N}(0,\ \sigma^2)\right) & \text{Uncalibrated noise.} \end{cases} \tag{7}$$

Ensuring that noise is calibrated is more complicated than detecting incorrect clipping because the gradient norm bound appears as a factor in both operations. Indeed, the gradient norm bound $C$ appears in Equation 7 in the term corresponding to clipping but also in the term injecting noise. We thus cannot simply vary the values for $C$ as we would not be able to separate the effect of $C$ on clipping from its effect on noise injection. Therefore, we first need to isolate the effect of the gradient norm bound $C$ on noise injection and eliminate its effect on clipping. To do so, we set the value of $C$ to be arbitrarily large so that the gradient is always left unclipped. In this way, $C$ no longer comes in as a factor in the first term of the noised gradient computation (for both uncalibrated and calibrated noised gradients). Then, we vary the value of $C$ to test whether the injected noise is indeed sampled from a Gaussian distribution whose scale is $C\sigma$ rather than $\sigma$. Put another way, by working with large values of $C$, we reduce Equation 7 to the following:

$$\text{Private Gradient} = \begin{cases} \frac{1}{B}\left(\sum_b g_b + \mathcal{N}(0,\ (C\sigma)^2)\right) & \text{Calibrated noise,} \\ \frac{1}{B}\left(\sum_b g_b + \mathcal{N}(0,\ \sigma^2)\right) & \text{Uncalibrated noise.} \end{cases} \tag{8}$$

There is an additional difficulty we need to overcome in order to distinguish these two Gaussian noises (of scale $C\sigma$ and $\sigma$). Recall from Section 3.1 that we cannot directly observe private gradients computed by the training algorithm. Instead, we would like to compare changes in the loss achieved by successive models outputted by the training algorithm. Because we are now studying a stochastic computation (i.e., adding noise to gradients), the outputs of DP-SGD can no longer be directly compared. To understand the reason for this state, it is easy to see how repeating the same training step without changing the gradient norm bound could lead to a different Gaussian sample being added to the gradient, which then cause different changes in the loss values. However, we can turn to statistical testing to address this difficulty.

**Theorem 1.** *Let $M_1, M_2$ be a pair of models that are trained with DP-SGD using the same initialization $M_0$ and the same mini-batch of data points $D_B$. Let us assume that the noise is sampled independently from the same Gaussian distribution $\mathcal{N}(0, s^2\mathbb{1}_K)$ and added to their gradients $G_1$ and $G_2$, where $\mathbb{1}_K$ is the identity matrix and $K$ is the dimension of the model's parameters. The parameter-space $l_2$-distance of $M_1$ and $M_2$ depends on the scale of the distribution of the noise $s$ added to their gradients:*

$$\mathbb{E}[\|M_1 - M_2\|_2] \propto s \tag{9}$$

*Proof.* Without loss of generality, we assume the aggregation method for the optimizer is to take the mean over the mini-batch, and that the optimizer is SGD. After one iteration of DP-SGD:

$$G_1 = G + \mathcal{N}(0, s^2\mathbb{1}_K), \quad G_2 = G + \mathcal{N}(0, s^2\mathbb{1}_K), \tag{10}$$

where $G$ is the aggregated per-example clipped gradients calculated using $D_B$.

Therefore, $\Delta G = G_1 - G_2 \sim 2\mathcal{N}(0, s^2\mathbb{1}_K)$.

$\|\Delta G\|_2^2 = \sum_k (\Delta G_k)^2 \sim \sum_k 2s^2\chi_1^2 \sim 2s^2\chi_K^2$, where $\chi_K^2$ is the chi-squared distribution with degree of freedom $K$. $\|\Delta G\|_2 = \sqrt{\|\Delta G\|_2^2} \sim \sqrt{2}s\chi_K$, where $\chi_K$ is the chi distribution with degree of freedom $K$. That gives $\mathbb{E}[\|\Delta G\|_2] = \sqrt{2}s\mathbb{E}[\chi_K]$.

Therefore, given $M_1 = M_0 - \eta G_1$ and $M_2 = M_0 - \eta G_2$, $\mathbb{E}[\|M_1 - M_2\|] = \mathbb{E}[\eta\|G_1 - G_2\|] \propto s$.

That is stating that if we repeat the DP-SGD training script multiple times and obtain multiple models $M_i$'s, the parameter-space $l_2$-distance between each pair of the models would have an expected value that is dependent on the scale of the noise. Therefore, by having multiple trained models, we are able to empirically estimate the expected value for the model distance by taking the mean over the model distance values of each pair of models. $\qquad\square$

We can substitute $s = C\sigma$ for the calibrated noise and $s = \sigma$ for the uncalibrated noise. This implies that if we fix the value for the noise multiplier $\sigma$, the expected value for the parameter-space $l_2$-distance between models trained with DP-SGD should be dependent on $C$ for the calibrated noise. Conversely, the expected distance should be independent from $C$ when the noise is not calibrated. This gives us a test that we can use to distinguish calibrated and uncalibrated noised gradient computations[2].

The detailed procedure of our test is as follows: (1) the developer first selects a model architecture and a set of data points (that could be real or synthetic); (2) the developer then picks a range of values for the gradient norm bound, $C \in [C_1, C_2, \dots]$ (that ensures the gradients are not clipped) as well as the noise multiplier $\sigma$. Heuristics for selecting $C$ and $\sigma$ are discussed in detail in Appendix B. Then (3) the developer runs the train script multiple (*e.g.,* 5) times with $C = C_1$ for a few iterations (*e.g.,* $T = 10$) and stores the final checkpoint received at the end of each training run $M_1, M_2, \dots, M_5$. They then compute the parameter-space distance between each pair of the models; (4) repeat step 3 for the rest of the values chosen for $C$; (5) plot the pairwise parameter-space distances with respect to $C$; and finally (6) compute the slope and run a regression $t$-test for the slope with null hypothesis of slope = 0. If the p-value is small (*e.g.,* p-value $\ll 0.05$), we can reject the null hypothesis and claim the noise is calibrated. Also, to ensure there is no false positive (i.e., the non-zero slope is not caused by calibrated noise, but rather an effect of clipping), the developer should repeat steps 3 to 5 with $\sigma = 0$ to ensure the slope of pairwise distance versus $C$ for the $\sigma = 0$ case is zero, which suggests that the $C$ values chosen have a minimal effect on training.

## 4 VALIDATION

Our tests are designed to identify mistakes in the DP-SGD implementation. Thus far, DP-SGD has been used in the vision (Abadi et al., 2016) and text (Dupuy et al., 2022) domains. As we demonstrate in this section, our proposed method can detect incorrect gradient clipping and uncalibrated noise addition in these domains. We consider three common models (ResNet20 (He et al., 2015b), WideResNet50 (Zagoruyko & Komodakis, 2016), and BERT (Devlin et al., 2019)) and three datasets (CIFAR-10 (Krizhevsky, 2009), ImageNet (He et al., 2015a), and SST2 (Socher et al., 2013)) (for implementation details, see Appendix F). In addition to DP-SGD, we also evaluate our method on DP-Adam to demonstrate its general applicability. Indeed, the Adam optimizer leverages additional techniques like momentum and per-parameter learning (Kingma & Ba, 2015). All of the experiments are repeated 5 times on 5 different machines to obtain a confidence interval. We analyze the computational cost for the proposed methods in Appendix D.

**A note on Adam.** We need to pay special attention when the Adam optimizer is used for training because it normalizes private gradients before they are applied to update the model. For the first training step of Adam, the private gradient would always be normalized by itself since the stateful optimizer is initialized with states of 0. This would result in similar model updates regardless of the difference in private gradients, making it hard to observe any differences between clipping cases

---

[2]Note that we do not assume we have access to the raw/private gradients. Thus, we are not able to directly use a statistical test to check for the distribution of the additive noise. See details in Appendix C.

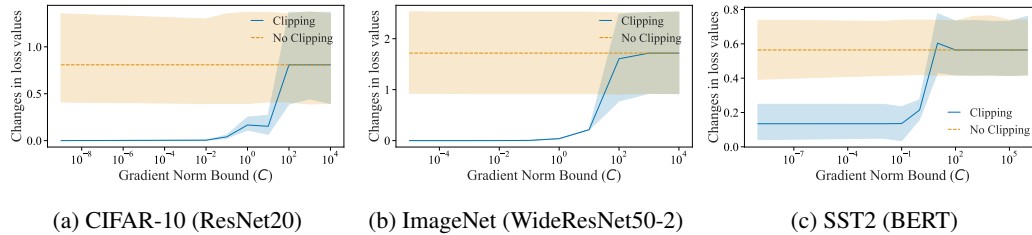

(a) CIFAR-10 (ResNet20)      (b) ImageNet (WideResNet50-2)      (c) SST2 (BERT)

Figure 2: Detecting the absence of gradient clipping. We train a model for 1 step to compute the change in the loss values and plot it with respect to the gradient norm bound while the noise addition is turned off (by setting $\sigma = 0$). A clear difference can be observed between the two curves in all the subplots: the loss change remains constant when there is no clipping, whereas it varies a lot when clipping is applied.

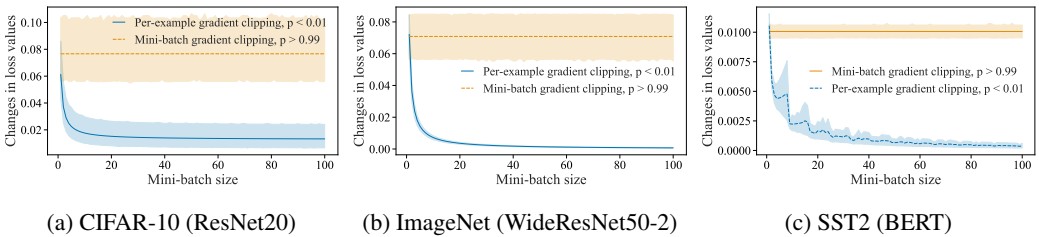

(a) CIFAR-10 (ResNet20)      (b) ImageNet (WideResNet50-2)      (c) SST2 (BERT)

Figure 3: Differentiating per-example gradient clipping versus mini-batch gradient clipping. We synthesized a mini-batch of zero-gradient data points along with one large-gradient data point, and then compute the loss changes when the model is trained on these data points while increasing the mini-batch size. A regression $t$-test is applied to each curve to test whether the slope is $0$ ($p$-values are included in the legends). It is consistently shown that the per-example gradient clipping leads to $p$-values that are always smaller than the $0.01$ significance level, whereas the $p$-values of mini-batch gradient clipping are always larger than $0.99$.

discussed in Section 3.1. To mitigate this problem and make loss comparison meaningful again, we train the model for a few iterations to make the states of the Adam optimizer non-zero. Another issue with Adam is that as the additive noise eventually begins to dominate the real gradient signal, the private gradient and its running momentum will be indistinguishable from noise. To prevent this failure mode, we work with small Gaussian scales when testing DP-Adam implementations. More details on the nuance of DP-Adam is provided in Appendix E

## 4.1 OUR APPROACH DETECTS INCORRECT GRADIENT CLIPPING IN DP-SGD IMPLEMENTATIONS

We first present empirical results demonstrating the effectiveness of our proposed method when it comes to debugging the correctness of a gradient clipping implementation. Unless otherwise specified, the noise multiplier is set to 0 for all the experiments in this subsection to eliminate the impact of noise added to the gradient.

**Clipping versus no clipping.** To verify if gradient clipping is implemented in a training script or not, we use the script to train the model for one single step with different gradient norm bounds, and compute the change in the loss value caused by this training step. Figure 2 (and Figure 5 in Appendix G for the Adam optimizer), show that changes in loss values are independent of $C$ if there is no clipping. In contrast, in the presence of clipping, the loss changes vary until $C$ is larger than the norm of the gradient. At this point, clipping is no longer applied: the two curves overlap with each other. Therefore, our test successfully detects the lack of gradient clipping in DP-SGD implementations. We set the mini-batch size to 1 so that per-example gradient clipping and mini-batch gradient clipping are equivalent.

**Per-example gradient clipping versus mini-batch gradient clipping.** After confirming that the DP-SGD implementation correctly uses some form of gradient clipping, the next logical step is to confirm that gradients are clipped on a per-example basis rather than at the level of the mini-batch aggregate. Recall from Section 3.1, when using our synthesized data points to train the model, the

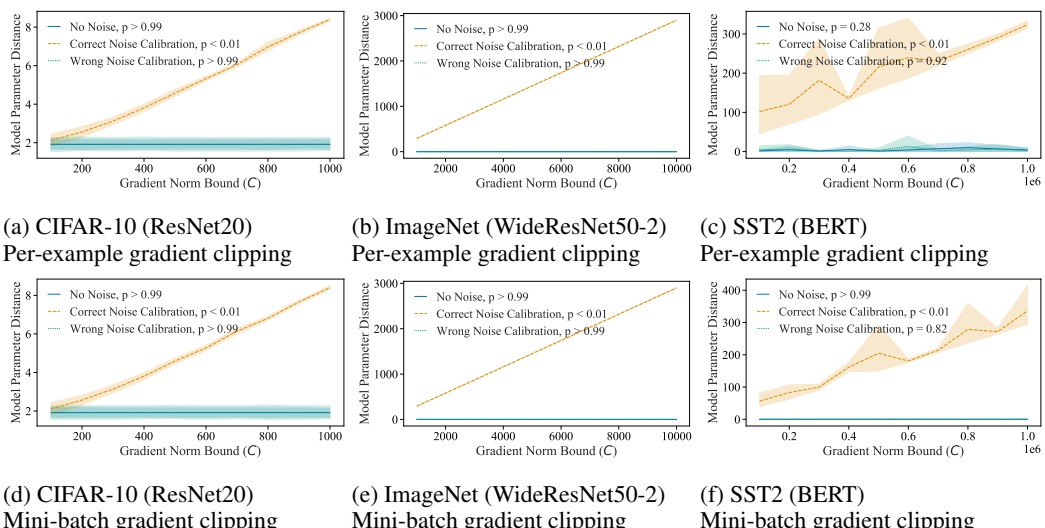

Figure 4: Verifying correctness of noise calibration for both per-example gradient clipping (first row) and mini-batch gradient clipping (second row) cases. We train multiple models for 10 steps and plot the pairwise model parameter distance with respect to the gradient norm bound for the three scenarios shown in the figure, respectively. $p$-values of a regression $t$-test with null hypothesis of 0 slope for the three curves are also reported. Observations: (1) model distances in both wrong noise calibration and no noise addition are independent of gradient norm bound; (2) the curve for correct noise calibration has a non-zero slope, which allows us to differentiate it with the wrong noise calibration; and (3) our proposed method is effective in detecting wrong noise calibration no matter whether the clipping is implemented correctly or not.

private gradients for mini-batch gradient clipping would be independent of the values of $B$. Yet, for per-example gradient clipping, the private gradients would depend on the mini-batch size. This is confirmed by Figure 3 for all three datasets where the changes in loss values are plotted with respect to different mini-batch sizes $(0, 100)$. We also report results for the Adam optimizer in Figure 6 in Appendix G. One can see that the amount of loss change in per-example gradient clipping decreases as the mini-batch size increases. This is intuitive given that magnitudes of the private gradients for the per-example gradient clipping case would decrease as $B$ increases, as shown in Equation 6. On the other hand, the private gradients for the mini-batch would stay constant with respect to $B$, hence the change in loss values would also stay constant. In conclusion, our test successfully differentiates mini-batch from proper per-example gradient clipping in DP-SGD implementations.

## 4.2 OUR APPROACH DETECTS UNCALIBRATED NOISE IN DP-SGD IMPLEMENTATIONS

To detect whether noise is calibrated according to the gradient norm bound $C$, we independently run the training script for 10 values for $C$. This should be repeated for two or more models for each value of $C$ in order to compute the pairwise parameter-space distance among the models for each $C$ as described in Section 3.2. In our experiments, we did this for five models to reduce uncertainty when reporting results in Figure 4 (first row). Note that besides the correct and wrong noise calibration, we also included a no-noise case (i.e., training with $\sigma = 0$) for reference. As expected, parameter distances are constant when the noise is uncalibrated (meaning the parameter distance is independent of $C$), and vice versa. To quantify the independence, we applied a regression $t$-test to check if the slopes of the curves are 0. It can be consistently seen across the three datasets that the slopes are non-zero with more than 99% confidence (i.e., $p$-value $\ll 0.01$) when the noise is correctly calibrated, whereas the $p$-values are large when the noise calibration is missing. We also evaluated the proposed method when the implementation of clipping is not correct (see Figure 4 d-f and Figure 7 in Appendix G) to show that our test of proper noise calibration does not rely on the correctness of clipping. In addition, we repeated this experiment on the Adam optimizer (see Figure 8 in Appendix G) and observed similar results, meaning our method applies to optimizers beyond DP-SGD.

## 5 CONCLUSION

In this work, we proposed a set of tests to debug implementations of DP-SGD. Unlike prior work, these tests are computational efficient and generally applicable. We are able to detect and identify two types of common mistakes that might occur during DP-SGD implementation, i.e., incorrect gradient clipping and improper noise calibration. Incorrectly computed private gradients are isolated based on an inspection of trained model behavior (*e.g.,* through parameter-space distance, and comparisons of loss values).

Related work investigated the vulnerabilities of DP training algorithms introduced by back-end software. For example, Jin et al. (2022) studied two threats that lead to side channel attacks. Both are explained by failed implementations of primitives that enable DP algorithms to sample noise. They demonstrated that these flaws are due to floating-point representations. However, these flaws are parallel to this work as they are not "bugs" caused by mistakes of the developers. These are indeed vulnerabilities introduced by libraries providing said primitives.

Our tests can be deployed without modifying existing training scripts: they only rely on accessing model checkpoints. We hope future work will extend our approach to debug DP guarantees of other algorithms such as Private Aggregation of Teacher Ensembles (PATE) (Papernot et al., 2017).

## ETHICAL IMPACT STATEMENT

Our work proposes tests that can be applied to debug implementations of DP-SGD, which is a training algorithms that is designed to protect differential privacy of training data points. Incorrectly implemented DP-SGD may lead to risks of privacy leakage. For example, one of the bugs that can be detected by our tests, mini-batch gradient clipping, can cause the privacy guarantees to be weaker by a factor that equals to the batch size, which is usually 128 times or more. While many ML developers start to take privacy into consideration, few of them are familiar with the implementation details of DP-SGD so that such bugs are common. Therefore, we believe research on debugging differentially private machine learning can solve this serious and urgent problem, and help developers to ensure and guarantee the privacy of their data providers, which is required by regulations such as the General Data Protection Regulation in the European Union, the California Consumer Privacy Act in the United States, and the Personal Information Protection and Electronic Documents Act in Canada.

## REPRODUCIBILITY STATEMENT

We will provide a link to an anonymous repository in the dicussion forums within the first week of the review process, and it will contain the code that can be used to reproduce our tests. After the review process, we will publicly open-source our code base along with documentations that explains how to use our tests and how to reproduce the experiments described in this paper.

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

## A COMPUTATION OF GRADIENT UPDATES

Using the data synthesizer, we get the following mini-batch of data points: $[0, 0, \ldots, 0, g]$ with mini-batch size of $B$, $||g||_2 \gg 0$. The clipping norm, $C$ is chosen such that $0 < C \ll \frac{||g||_2}{B}$. When we repeat this for a set of different mini-batch sizes $B_i$'s, $C$ should be chosen to be $\ll \frac{||g||_2}{\max B_i}$. With per-example gradient clipping, the batch of clipped gradients will be $[0, 0, \ldots, 0, \frac{C \cdot g}{||g||_2}]$ and the aggregate update would be the mean of the clipped gradients, $\frac{C \cdot g}{B \cdot ||g||_2}$. With mini-batch gradient clipping, the aggregate gradient update would be $\frac{g}{B}$. After clipping, the aggregate update would be $\frac{g/B}{||g/B||_2} \cdot C = \frac{C \cdot g}{||g||_2}$, which is independent of $B$.

## B HEURISTICS TO CHOOSE $C$ AND $\sigma$ FOR NOISE CALIBRATION DETECTION

As described in the Section 3.2, eliminate the effect of clipping when observing the noise distribution, we need to choose the values for $C$ that they are large enough such that the gradients are not clipped at all. In order to estimate the minimally required value for $C$, we performed the following steps:

1. For the chosen architecture and dataset, we trained a model from random initialization for $T$ steps (in our experiments, we used $T = 1$ when using the SGD optimizer and $T = 5$ when using Adam optimizer). Using this trained model as the initial state for all steps afterward.

2. Set $\sigma = 0$. Train the model with the DP-SGD implementation for $C \in [0.01, 0.1, 1.0, 10, 100, 1000, 10000, 100000, 1000000]$ for 1 step and compute the change in loss for each value of $C$.

3. Pick $C = C^*$ if the change of loss values stop changing for all $C_i > C^*$.

4. Repeat steps 1, 2, and 3 for multiple times and use the maximum of selected $C^*$ value from all trials.

5. Sample 10 values for $C \in [C^*, 10C^*]$

After selecting the minimal value of gradient norm bound, we will need to pick a reasonable value for $\sigma$. This is important because if $\sigma$ is too small the effect of additive noise might not be observable; yet if $\sigma$ is too big, (i.e., the additive noise dominates over the real gradient signal), this will cause optimizers like Adam to normalize every element of the gradient update to be +/-1. Therefore, in order to final the optimal value for $\sigma$ during the test, we propose to perform the following steps:

1. Using the model state generated after the first step when testing for optimal $C$ value.

2. Set $C = C^*$. Train the model with the DP-SGD implementation for $\sigma \in [1 \times 10^{-8}, 1 \times 10^{-7}, 1 \times 10^{-6}, \ldots, 1.0, 10.0]$ for 1 step and compute the change in loss for each value of $\sigma$.

3. Pick $\sigma = \sigma^*$ if the change of loss values starts to change for all $\sigma_i > \sigma^*$.

4. Repeat steps 1, 2, and 3 for multiple times and use the minimum of selected $\sigma^*$ value from all trials.

## C KS TEST

It should be noted that the Kolmogorov–Smirnov (KS) test could be used to detect noise calibration if the optimizer is only vanilla SGD. KS test is a non-parametric test that can estimate if the two distributions are the same. We can train two models with same initial weight $W_0$ and their own copy of a DP-SGD optimizer initialized with the same state, i.e., momentum term would be zero for the first step of SGD. For the first model, we train one step to get $M_1$ with noise multiplier of 0, i.e., no noise, with a mini-batch with the size of $B$ and a learning rate of $\eta$. For the second model with non-zero noise (i.e., $\sigma \neq 0$) and the same $B$, $\eta$, and the data points to reach $M_2$. In this setting, $M_2 - M_1$ would be exactly the noise we added (scaled by $B$ and $\eta$). For the correctly calibrated

noise, $M_2 - M_1 \sim \frac{\eta}{B}\mathcal{N}(0, (C\sigma)^2)$. We can run a KS test to see if $M_1 - M_2$ follows the expected distribution and scale. However, this approach only works on a SGD optimizer. For another type of optimizer (i.e., Adam), the model update includes other calculations such as normalization. In this case, we are no longer capable of using a KS test to detect the distribution of the additive noise.

## D  COMPUTATIONAL COST

**Check for Existence of Clipping.**  The total number of steps required to complete the test is # training steps $\times$ # Gradient Norm Bound. We used 1 training steps for 8 gradient norm bounds so the total cost is equal to training for 8 steps. For reference, CIFAR-10 with batch size 100 needs 500 steps per epoch. So the cost of checking is minimal.

**Check for Per-Example Gradient Clipping.** The total number of steps required to complete the test is # training steps $\times$ # mini-batch sizes. We use 1 training step and 100 different batch sizes. So the cost is similar to 100 steps of normal training (i.e., 0.2 epochs of normal training).

**Check Noise Calibration.** The total number of steps required to complete the test is # models $\times$ # training steps per model $\times$ # Gradient Norm Bound, *e.g.,* $2 \times 10 \times 10 = 200$ steps (i.e., 0.4 epochs of normal training).

## E  MORE ON DP-ADAM

Adam optimizer can be thought of as a combination of Momentum and RMSprop. It is also one of the most popular optimizers used in deep learning training. Hence we believe studying the effectiveness of our proposed method on Adam demonstrates it is generally applicable to different variants of SGD optimizer.

Adam optimizer has two state parameters, first momentum $m_t$ and second momentum $v_t$ that keep track of the running history of $g_t$ and $g_t^2$ respectively, where $g_t$ is the gradient (i.e., $m_t = \beta_1 \cdot m_{t-1} + (1 - \beta_1)g_t$; $v_t = \beta_2 \cdot v_{t-1} + (1 - \beta_2)g_t^2$). The model update would be $-\eta \cdot m_t/\sqrt{v_t + \epsilon}$ with some small $\epsilon$ for numerical stability. (Without loss of generality, we omit the bias correction for the first and second momentum terms here for the point of demonstration. The conclusion stays the same.)

In the DP implementation, the raw gradient is clipped and noised, then the private gradient is passed into the optimizer as $g_t$. Therefore, the first training step of Adam, when $m_0$ and $v_0$ are initialized to be 0, $m_1 = g_1$ and $\sqrt{v_1 + \epsilon} \approx g_1$. It means the private gradient is normalized by itself for the first training step. Therefore, if we only train for 1 step, the model update would look similar regardless of the difference in $g_1$. The normalized gradient would be a vector of $\pm 1$'s. In this case, the distance between the models does not reflect the scale of the additive noise. But if we train for multiple steps, the state parameters will keep track of the running history of the private gradients $g_t$ (which consist of both the raw gradients and the additive noise). For any later steps, e.g., at step 2, the $m_2 = \beta_1 \cdot m_1 + (1 - \beta_1)g_2$; $v_2 = \beta_2 \cdot v_1 + (1 - \beta_2)g_2^2$). The noise term in $g_t$ in update $m_2/\sqrt{v_2 + \epsilon}$ cannot be simplified as that for the first step, hence the resultant model updates would be dependent on the additive noise. Even though the model distance may not strictly be proportional to the scale of the noise after this normalization, but its general dependency on the noise scale still holds. Therefore, when using DP-Adam, we need to run multiple steps to observe any useful trend.

Another observation about DP-Adam is that, when $C$ is large, the model parameter distance saturates/plateaus (as seen in Figure 8 in Appendix G). We believe it is because the scale of calibrated noise also increases as we increase $C$. Eventually, the noise would dominate the real gradient signal, causing the private gradient and running momentum to be both indistinguishable from the noise. In this case, the normalized gradient updates would all look "similar" even if we keep increasing the scale of the noise (by increasing $C$) and the distance between them would saturate.

## F  IMPLEMENTATION DETAILS

In our code, both our tests and the training scripts of the developers are written by using the PyTorch library (Paszke et al., 2019). Regarding model architectures, we use ResNet20 from `github.com/`

`akamaster/pytorch_resnet_cifar10` (Idelbayev), WideResNet50-2 from the Torchvision library, and BERT from the Transformers library (Wolf et al., 2019). We manually replaced the batch normalization layers of ResNet20 and WideResNet50-2 with group normalization layers because the former is not compatible with differentially private training. We used the Opacus library (Yousefpour et al., 2021) for correct DP-SGD, whereas the DP-SGD with mistakes is written by ourselves using the PyTorch library or modified from Opacus library.

## G ADDITIONAL FIGURES

Figure 5 and Figure 6 are additional evaluation results for Section 4.1. Figure 7 and Figure 8 are additional results for Section 4.2.

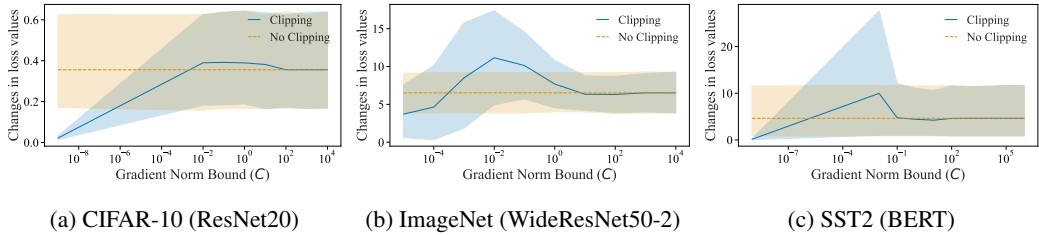

| (a) CIFAR-10 (ResNet20) | (b) ImageNet (WideResNet50-2) | (c) SST2 (BERT) |

Figure 5: This is the reproduction of Figure 2 except the optimizer used here is the Adam optimizer. Similar results can be seen so we believe the proposed method is effective across different optimizers.

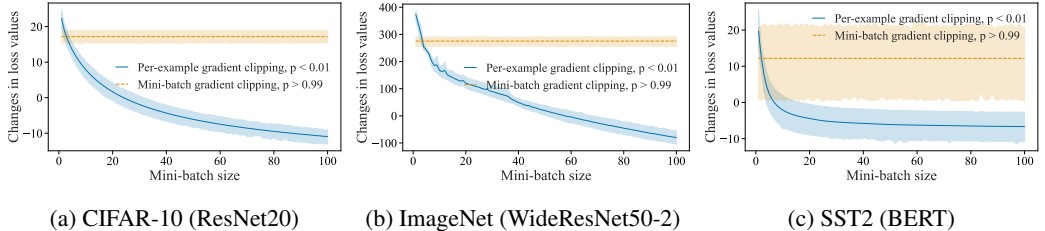

| (a) CIFAR-10 (ResNet20) | (b) ImageNet (WideResNet50-2) | (c) SST2 (BERT) |

Figure 6: This is the reproduction of Figure 3 except the optimizer used here is the Adam optimizer. Similar results can be seen so we believe the proposed method is effective across different optimizers.

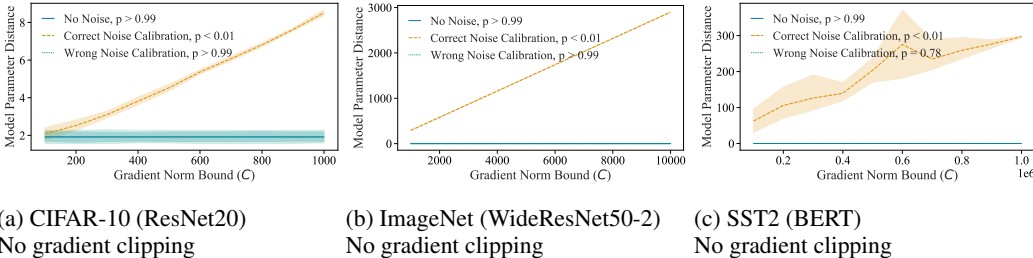

(a) CIFAR-10 (ResNet20)
No gradient clipping

(b) ImageNet (WideResNet50-2)
No gradient clipping

(c) SST2 (BERT)
No gradient clipping

Figure 7: This is the a reproduction of Figure 4 except we consider the case of no gradient clipping. Similar results can be seen so we believe the proposed method is effective no matter whether the clipping is implemented correctly or not.

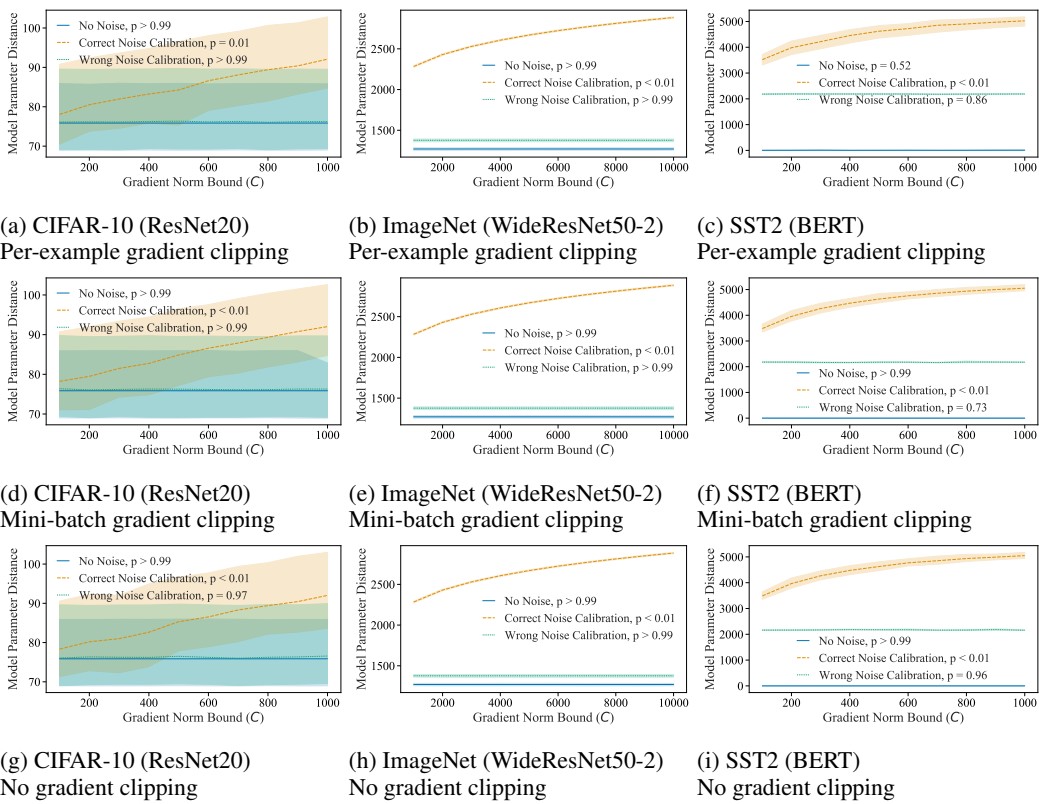

(a) CIFAR-10 (ResNet20)
Per-example gradient clipping

(b) ImageNet (WideResNet50-2)
Per-example gradient clipping

(c) SST2 (BERT)
Per-example gradient clipping

(d) CIFAR-10 (ResNet20)
Mini-batch gradient clipping

(e) ImageNet (WideResNet50-2)
Mini-batch gradient clipping

(f) SST2 (BERT)
Mini-batch gradient clipping

(g) CIFAR-10 (ResNet20)
No gradient clipping

(h) ImageNet (WideResNet50-2)
No gradient clipping

(i) SST2 (BERT)
No gradient clipping

Figure 8: This is the a reproduction of Figure 4 and Figure 7 except that the optimizer used here is the Adam optimizer. Similar results can be seen so we believe that the proposed method is effective across different optimizers.

