# OpenReview forum: "Finding Private Bugs: Debugging Implementations of Differentially Private Stochastic Gradient Descent "
_ICLR.cc/2023/Conference — Submitted to ICLR 2023_

### Official Review · Reviewer_Tpks · 2022-10-20

**Confidence:** 4
**Correctness:** 4
**Technical Novelty And Significance:** 2
**Empirical Novelty And Significance:** 2
**Recommendation:** 3

**Clarity, Quality, Novelty And Reproducibility:**

The paper is clear and well-written. The proposed approach is novel, although limited in scope. I believe the main results from the paper are easy-enough to reproduce.


**Strength And Weaknesses:**

### Strengths:

1) The paper is generally very easy to follow and nice to read.

2) The proposed approach seems solid.

3) Debugging DP software can be really painful, since often the main signal of potential problems is that everything works too well. Good debugging tools are potentially very helpful.

### Weaknesses:

1) My main (and pretty much only) problem with the paper is that the current problems it aims to solve are quite specific (only DP-SGD) and not too hard to diagnose without the proposed tools. I would be more enthusiastic if the authors can expand the work to provide a framework for debugging DP software (at least somewhat) more generally. I do not believe that the current work warrants a publication in ICLR.



**Summary Of The Paper:**

The paper introduces light-weight approaches for debugging differentially private stochastic gradient descent (DP-SGD). The authors first identify the possible bugs which invalidate DP-SGD privacy guarantees (per-example clipping, noise calibration), and then proposes tests that aim to identify the exact problem (no gradient clipping, not clipping per-example grads; no noise, wrong noise scaling) without having to modify the source code. The authors empirically test their proposed solution to show that it correctly identifies various implementation issues in DP-SGD.


**Summary Of The Review:**

A nice paper to read, but the current problem is too simple and lacks ambition.


### Update after rebuttal

After reading the other reviews and the author responses, I keep my score unchanged and recommend rejection: as I commented to the authors, I think debugging DP-SGD is simply not important enough to warrant publication in ICLR. If I would be to review this paper again in the future, the most important update that would persuade me to increase my score would be to try and make the debugging tools more widely applicable.

---

> ### Author Response · Authors · 2022-11-15
> **Response**
>
> We thank the reviewer for their comments.
>
> > **My main (and pretty much only) problem with the paper is that the current problems it aims to solve are quite specific (only DP-SGD) and not too hard to diagnose without the proposed tools. I would be more enthusiastic if the authors can expand the work to provide a framework for debugging DP software (at least somewhat) more generally.**
>
> We would like to clarify that differential privacy is defined on the algorithmic level, it is important to tailor tests to the specifics of the training algorithm. This means it is not possible to introduce generic tests for multiple DP training algorithms. We are aware that there are studies on testing the privacy of trained models, for example, by applying membership inference attacks. However, it is worth noting that robustness against such privacy attacks does not provide any indication of  the training algorithm providing differential privacy guarantees. Besides, such methods usually test for privacy leakage from a specific model whereas our testing approach evaluates the training algorithm itself - such that any model that is trained using this implementation possesses privacy advantages. Therefore, we believe our work is adjacent to such methods, and much more advantageous in the scenario of a developer seeking to validate the correctness of their implementation of privacy-preserving training.

---

> > ### Comment · Reviewer_Tpks · 2022-12-02
> > **Thanks for the rebuttal**
> >
> > I am aware that attacking models can only show if DP does not hold, and that just checking that an arbitrary algorithm is DP is computationally infeasible. What I was hoping for was a debugging tool with a more limited scope (think of other standard DP mechanisms, like output perturbation or objective perturbation, or even more special cases like sufficient statistics perturbation), but that goes beyond DP-SGD. Unfortunately, I am currently not convinced that the main contribution in the paper is enough to warrant publication in ICLR.

---

### Official Review · Reviewer_y7ow · 2022-10-23

**Confidence:** 3
**Correctness:** 3
**Technical Novelty And Significance:** 4
**Empirical Novelty And Significance:** 4
**Recommendation:** 6

**Clarity, Quality, Novelty And Reproducibility:**

# Clarity

- The paper is overall clear and one understands the main message.
- However, I think that the tests could be formalized a bit more, in particular regarding the assumptions that are made on the black box. This would make it easier to understand the conditions of validity of the proposed tests.
  - An example of such a constraint is in Sec 3.1,
>Note that we would like our test to be implementable without having to make modification to the training script

  I think that summarizing all these assumptions in a dedicated section would help clarify the paper
  - Another example is the level of control the developer has on the samples provided to the black-box: is it controllable? Some sections hint at a positive answer, but I would encourage the authors to explicit it.
  - In sec 3.1, to detect mini-batch gradient clipping, the test relies on creating a specific batch of size $B$ such that $B-1$ samples have zero gradient and $B$ has a large gradient. If $\ell(\hat{y}, y)$ is the loss between the predicted $\hat{y} = M(x)$ and the true label $y$, the method proposed by the paper implicitly assumes that $\nabla_{\hat{y}} \ell (\hat{y}, \hat{y}) = 0$. This holds for the cross-entropy and the mean square error, but not necessarily for arbitrary losses.
- I think the abstract and the conclusion are slightly misleading as regards the mistakes actually covered by the paper. The sentences
> These mistakes include improper gradient computations or noise miscalibration

  (abstract) and
  > We are able to detect and identify common mistakes like incorrect gradient clipping and improper noise calibration

  (conclusion) wrongly give the impression that gradient computations and noise calibration are a *strict subset* of the properties of DP-SGD tested here. In fact, they are the only two. I think clarifying this contribution (which is perfectly valid on its own) would be less misleading regarding the actual paper content.

## Minor typos and comments
- Weights of the model are denoted $W$ L8 of Algorithm 1, but actually written $M$ in Eq (3) and other main text references
- In Eq (6), a subscript $g_b$ is missing to $g$
- In the proof of theorem 1, the last paragraph should probably not be included in the proof itself but after (QED symbol ill placed)

# Quality

- The proposed tests for the gradient clipping (i) are sound to me in the case of DP-SGD.
- Similarly, the proposed tests for noise calibration (ii), although slightly less clear than Sec 3.2, also seem relevant for DP-SGD
  - However, I don't understand why there is necessarily a need to directly look at the models; couldn't one use the losses, as in the case (i)? The need for performing long training steps ($100$ updates) is not clear to me either. For instance, for DP-SGD, for a fixed data batch and weights, one could do multiple single steps for different values of $C$ and check if the loss variation's variance depends on $C$. This question is also related to what the experimenter has access to / preferably accesses with respect to the model training loop.
- The validity of the tests proposed to address (ii) in the case of DP-Adam is not entirely clear to me. Indeed, Adam implies a renormalization, so it is unclear why the results should depend on $C$. For instance, in Fig 8 in Appendix E, one sees that the distance between models saturate: is this result expected?

# Novelty

The proposed approaches are novel to the best of my knowledge.

# Reproducibility

- In sec 3.1, the authors introduce a parameter $\alpha \gg 1$ to ensure having a "large enough" gradient. How is chosen $\alpha$?


**Details Of Ethics Concerns:**

No issue

**Strength And Weaknesses:**

# Strengths

- (Impact) This paper will be appealing for DP practitioners, as it addresses a real need to ensure the privacy of actual implementations of DP. I think this goes in a much-needed direction.
- The proposed tests are simple conceptually, yet effective for (i) and (ii), lightweight, and can be plugged over any black-box implementation of DP (agnosticism)

# Weaknesses

- (Clarity) The paper would benefit from a more formal writing: some hypotheses are buried within the text, others are missing, and the theoretical justification of the noise calibration test for DP-Adam lacks rigor (see more detailed comments below).
- (Limitations) A more minor weakness is that not all potential pain issues of DP implementations are covered, notably batch sampling issues: the paper is limited to noise calibration and gradient clipping

**Summary Of The Paper:**

This paper investigates the problem of finding out whether an implementation of DP-SGD is subject to silent bugs affecting the privacy protection, focusing on (i) gradient clipping and (ii) noise scaling. For each case, it proposes simple tests that relying on varying DP hyperparameters (batch size, noise level, clipping bound). The tests only assume the knowledge of a model training function function (black-box), which makes them agnostic and generic. The resulting tests are empirically validated on several datasets and architectures.

**Summary Of The Review:**

This paper addresses a practical problem of large importance and proposes simple yet effective methods. However, the paper could benefit from an improvement in writing, and the applicability of the tests to DP-Adam would require more theoretical justifications. If the authors addressed these concerns, I would be willing to significantly increase my score, as I think this paper could be impactful for the community.

---

> ### Author Response · Authors · 2022-11-15
> **Response 1**
>
> We thank the reviewer for their careful reading and detailed comments.
>
> > **A more minor weakness is that not all potential pain issues of DP implementations are covered, notably batch sampling issues: the paper is limited to noise calibration and gradient clipping**
>
> Thank you for the suggestion. In our work, we focus on modifications that DP-SGD imposes to vanilla SGD, not those that are common between SGD and DP-SGD (such as uniform data sampling). This is because developers, especially those who do not have DP expertise, may forget to incorporate or incorrectly implement  these modifications as developers are used to SGD, for more details see the last 2 paragraphs in Section 2:
> “For example, developers are used to computing gradients for mini-batches of training examples when implementing vanilla SGD as opposed to per-example gradients, i.e., developers may tend to only obtain the aggregated gradient update. This course of action can cause mistakes in the implementation of per-example clipping by either performing the mini-batch aggregation before gradient clipping or missing the clipping altogether. In addition to this, developers, especially those who do not have DP expertise, may forget to scale the noise by C thus implementing uncalibrated noise, thereby rendering the calculation of $\epsilon$ independent of C.”
>
> > **However, I think that the tests could be formalized a bit more, in particular regarding the assumptions that are made on the black box. This would make it easier to understand the conditions of validity of the proposed tests. An example of such a constraint is in Sec 3.1,
> Note that we would like our test to be implementable without having to make modification to the training script. I think that summarizing all these assumptions in a dedicated section would help clarify the paper. Another example is the level of control the developer has on the samples provided to the black-box: is it controllable? Some sections hint at a positive answer, but I would encourage the authors to explicit it.**
>
> Thank you for your thoughtful comment. We have three main assumptions (added in Section 3):
>
> 1. No real data: the developer should not need real data points for running tests, they should also be able to use synthetic data points.
> 2. No privacy risk: any gradient computation or model trained as part of the proposed tests does not need to be released; put another way the sole purpose of these tests is to assess the correctness of a DP-SGD implementation. This means the approach does not increase the risk of leakage of private information when these tests were executed using sensitive inputs from real data.
> 3. No changes: tests should not impose any changes to training scripts, and they should be easy and light to run.
>
>
> > **In sec 3.1, to detect mini-batch gradient clipping, the test relies on creating a specific batch of size B such that B−1 samples have zero gradient and B has a large gradient. If ℓ(y^,y) is the loss between the predicted y^=M(x) and the true label y, the method proposed by the paper implicitly assumes that ∇y^ℓ(y^,y^)=0. This holds for the cross-entropy and the mean square error, but not necessarily for arbitrary losses.**
>
> Thanks for pointing this out. Since the goal here is to minimize the magnitude of gradients through a careful choice of label that minimizes the loss, we will generalize our method to setting y = argmin_y ℓ(y^,y) without modifying other parts of the test. Since the loss function is a mapping from the model’s output space (which is usually low-dimensional) to a scalar, solving y anaytically or numerically should be tractable. For the cross-entropy and the mean square error, the solution would be y=y^. We added this to Section 3.1.
>
> We would like to clarify that the only requirement of our proposed method for detecting mini-batch gradient clipping is to obtain a data point that leads to a large gradient (such that it would be clipped) and another data point that leads to a small gradient so that it causes a negligible impact on the change in loss values. With the ability to modify the data points, this is typically not hard to achieve.
>
> > **abstract & conclusion are misleading. clarifying the contribution**
>
> We have rephrased those parts.

---

> > ### Author Response · Authors · 2022-11-15
> > **Response 2**
> >
> > > **However, I don't understand why there is necessarily a need to directly look at the models; couldn't one use the losses, as in the case (i)?**
> >
> > We found that changes in the loss do not reliably detect miscalibrated noise addition, especially when the Adam optimizer is used. This is confirmed by this figure (https://anonymous.4open.science/r/ICLR-17B1/change_in_loss_noise_calibration.pdf),  where we plotted the change in loss v.s. the gradient norm bound C for both wrong noise calibration (orange) and correct noise calibration (blue).
> > We hypothesize that this is because, unlike parameter distance, a change in the loss does not have a near-linear relation to the noise multiplier. In other words, adding a larger noise does not necessarily lead to a larger/smaller change in loss. Also recall that the noise multiplier is set to 0 (i.e., we disable noise addition) for the two tests of clipping so changes in the loss can be used there, but of course we cannot do the same for checking noise calibration. We will add a footnote about this to the paper if you think this helps understanding.
> >
> >
> > > **The need for performing long training steps (100 updates) is not clear to me either.**
> >
> > Thank you for the question. For the SGD optimizer, it is not necessary to train for multiple steps. This is only required for the Adam optimizer. Hence, for consistency of the method, we propose to train for multiple steps for both of the optimizers. We briefly discussed nuances introduced by the DP-Adam optimizer in Section 4. We would like to elaborate a little bit on that.
> >
> > First of all, the Adam optimizer can be thought of as a combination of Momentum and RMSprop. Hence we believe studying the effectiveness of our proposed method on Adam demonstrates it is generally applicable to different variants of SGD optimizer.
> >
> > Recall that the Adam optimizer has two state parameters, first momentum $m_t$ and second momentum $v_t$, that keep track of the running history of $g_t$ and $g_t^2$ respectively, where $g_t$ is the gradient (i.e., $m_t = \beta_1 \cdot m_{t-1} + (1-\beta_1) g_t ;  v_t = \beta_2 \cdot v_{t-1} + (1-\beta_2) g_t^2)$. The model update would be $-\eta \cdot m_t / \sqrt{v_t + \epsilon}$ with some small $\epsilon$ for numerical stability. (Without loss of generality, we omit the bias correction for the first and second momentum terms here.)
> >
> > In the DP implementation, the raw gradient is clipped and noised, then the private gradient is passed into the optimizer as $g_t$. Therefore, the first training step of Adam, when $m_0$ and $v_0$ are initialized to be $0$, $m_1 = g_1$ and $\sqrt{v_1 + \epsilon} \approx g_1$.  It means the private gradient is normalized by itself for the first training step. Therefore, if we only train for 1 step, the model update would look similar regardless of the difference in $g_1$. The normalized gradient would be a vector of $\pm 1$’s. In this case, the distance between the models does not reflect the scale of the additive noise. But if we train for multiple steps, the state parameters will keep track of the running history of the private gradients $g_t$ (which consist of both the raw gradients and the additive noise). For any later steps,  e.g., at step 2, the $m_2 = \beta_1 \cdot m_1 + (1-\beta_1) g_2 ; v_2 = \beta_2 \cdot v_1 + (1-\beta_2) g_2^2)$. The noise term in $g_t$ in update $m_2 / \sqrt{v_2 + \epsilon}$ cannot be simplified as that for the first step, hence the resultant model updates would be dependent on the additive noise. Even though the model distance is not necessarily strictly proportional to the scale of the noise after this normalization, its general dependency on the noise scale still holds. Therefore, when using DP-Adam, we need to run multiple steps to observe any useful trend.
> >
> > Having justified why we consider multiple training steps, we now ask if we can require fewer training steps. We find in additional experiments we ran following the reviewer’s comment that it is enough to train with fewer steps (in particular, in the order of 10). We have updated our experiments using 10 training steps for Figure 4 in Section 4.2 and additional figures (fig. 7 and 8) in Appendix G. How should developers choose the number of steps? We recommend they use the p-value as an indicator of when to stop requiring additional steps. If the number of steps is not large enough (e.g., less than 10 in our setup), we observe drops in the p-values of some incorrect implementations of DP-Adam (e.g., the p-value for the no noise case on the SST2 dataset is 0.57). While they are still large enough to be detected, we discourage further reducing the training steps to avoid false negatives. We added the content above to Appendix E.

---

> > > ### Author Response · Authors · 2022-11-15
> > > **Response 3**
> > >
> > > > **The validity of the tests proposed to address (ii) in the case of DP-Adam is not entirely clear to me. Indeed, Adam implies a renormalization, so it is unclear why the results should depend on C. For instance, in Fig 8 in Appendix E, one sees that the distance between models saturate: is this result expected?**
> > >
> > > Since the raw gradient is clipped and noised, then the private gradient is passed into the optimizer as private gradient, $g_t$ in DP-Adam implementation, $g_t$ is a function of raw gradient and the additive noise (note we have omitted the effect of clipping by choosing large gradient norm bounds C). Even though for the first step, $m_1=g_1$ and $v_1=g_1^2$, the normalized gradient update would be a vector of $\pm 1$. But for later steps, e.g., at step 2, the $m_2 = \beta_1 \cdot m_1 + (1-\beta_1) g_2 ; v_2 = \beta_2 \cdot v_1 + (1-\beta_2) g_2^2)$. This time, the noise term in $g_t$ in update $m_2 / \sqrt{v_2 + \epsilon}$ cannot be simplified as that for the first step, hence the resulting model updates would be dependent on the additive noise. Even though the model distance is not necessarily strictly proportional to the scale of the noise after this normalization, its general dependency on the noise scale still holds.
> > >
> > > In terms of the saturation when C is large for DP-Adam, it is observed because the scale of calibrated noise also increases as we increase C in the case of correctly calibrated noise. Eventually, the noise would dominate the real gradient signal, causing the private gradient and running momentum to be both indistinguishable from the noise. In this case, the normalized gradient updates would all look "similar" even if we keep increasing the scale of the noise (by increasing C) and the distance between them would saturate.
> > >
> > > We have added the content above to Appendix E. Hope this clarifies the concerns on DP-Adam and the explanation .
> > >
> > > > **How to choose alpha?**
> > >
> > > Any alpha that causes a non-zero per-example gradient with a larger norm than $C$ (ensuring the presence of clipping) would work. We confirmed this point by performing experiments using  alpha of 10, 100, 1000 and observing similar results. In our paper, we picked alpha=10 and clarified this in Section 3.1.

---

> ### Comment · Reviewer_y7ow · 2022-12-04
> **Thanks for your answer**
>
> I thank the authors for their detailed answers which addresses most of my concerns. I slightly increased my score.
>
> Coming back to the issue of sampling, I still would like to point out that batch sampling in SGD is not necessarily the same as in DP-SGD if one wants valid DP bounds, see e.g. the need for the [uniform samplers](https://opacus.ai/api/utils/uniform_sampler.html) introduced in Opacus.

---

### Official Review · Reviewer_HFjU · 2022-10-27

**Confidence:** 3
**Correctness:** 3
**Technical Novelty And Significance:** 2
**Empirical Novelty And Significance:** 3
**Recommendation:** 6

**Clarity, Quality, Novelty And Reproducibility:**

The paper is clear and it has good quality. The novelty is mostly in the checking of the added noise. The results seem reproducible.

**Strength And Weaknesses:**

Strengths:

1. I think the problem of debugging DP implementations and having appropriate tests is really important, and practitioners could really benefit from built tools that could easily test their implementations.



Weaknesses:

1. I think as this paper is more on the development/coding side, it is really important to have the code but I could not find a link to the anonymized repository. In general, how the tests/checks are implemented, how easy they are to use and how efficient they are is super important, as the checks themselves are very easy to come up with. The first two, i.e. changing clipping factor and batch size is something that I have done before. Another common thing people check is changing the epsilon value and setting it to very high and seeing how the performance improves. The last check however is novel and interesting.

2. One thing that should be better clarified in the paper and treated with caution is that the first two checks should really not be done on actual private, eyes-off data! clipped gradients are still private and anything that is a function of them is off limits. So anything prior to noisy-ing the gradients cannot be seen. However, the good news is any data (even fake synthetic) would work here. I think this should be specified in the paper, otherwise there is privacy violations.

3. I worry about the efficiency of the method as for the checks things need to change and model needs to be trained multiple times. This could have a lot of overhead and it needs further analysis.

4. This is minor: I think the first 3 pages of the paper are very repetitive, without providing much information. This is especially the case for the intro. I think it would be better if the intro actually gave an overview of how you do the check, rather than repeating the stuff that is being checked.

**Summary Of The Paper:**

This paper introduces a practical dev-level method for debugging differentially private SGD (DP-SGD), so that practitioners can check their implementations, and make sure all the steps of DPSGD  are working correctly and that the guarantee is achieved. Unlike the direction of work that does checking by finding bounds through attacks, this work focuses on the implementation and the training. More specifically, the proposed method checks the following three things a) clipping is happening, b) clipping is happening on a per-example level rather than a minibatch level c) noise addition is being done correctly. They do the first two checks by looking at the difference in loss by changing the clipping factor (for a) and minibatch size (b). They do the last check by training models multiple times and finding the standard deviation between the final parameters of the different models (looking at the l2 distance between model parameters).

**Summary Of The Review:**

I like the idea of having implementation checks for DP-SGD. Although I find the first two checks a bit obvious, I find the third one novel and I think having an implementation of them that can be easily used is really important.

---

> ### Author Response · Authors · 2022-11-15
> **Response 1**
>
> We thank the reviewer for their thoughtful review.
>
> > **I think as this paper is more on the development/coding side, it is really important to have the code but I could not find a link to the anonymized repository. In general, how the tests/checks are implemented, how easy they are to use and how efficient they are is super important, as the checks themselves are very easy to come up with.**
>
> The code required to reproduce our experiments is available at https://anonymous.4open.science/r/Finding-Private-Bugs-4087
>
> > **One thing that should be better clarified in the paper and treated with caution is that the first two checks should really not be done on actual private, eyes-off data! clipped gradients are still private and anything that is a function of them is off limits. So anything prior to noisy-ing the gradients cannot be seen. However, the good news is any data (even fake synthetic) would work here. I think this should be specified in the paper, otherwise there is privacy violations.**
>
> Thanks for the suggestion. You are correct that our tests should not be executed using sensitive data if the developer is not allowed to inspect such data. A developer not able to use real data to run the tests could indeed default to synthetic data. Formally, we make the following assumptions on the developer’s access: (1) The developers have access to the training script and any intermediate model checkpoints it outputs. The developer is able to control the hyperparameters (e.g., mini-batch size, number of training steps, noise multiplier σ, Gradient Norm Bound C, etc.). (2) Any gradient computation or model trained as part of the proposed tests should not be released when these tests were executed using sensitive inputs from real data. This is not a limitation given that the sole purpose of these tests is to assess the correctness of a DP-SGD implementation. Alternatively, the developer can always choose to run the tests on synthetic data. We clarified this in the revised manuscript by adding three main assumptions in Section 3:
>
> 1. No real data: the developer should not need real data points for running tests, they should also be able to use synthetic data points.
> 2. No privacy risk: any gradient computation or model trained as part of the proposed tests does not need to be released; put another way the sole purpose of these tests is to assess the correctness of a DP-SGD implementation. This means the approach does not increase the risk of leakage of private information when these tests were executed using sensitive inputs from real data.
> 3. No changes: tests should not impose any changes to training scripts, and they should be easy and light to run.

---

> > ### Author Response · Authors · 2022-11-15
> > **Response 2**
> >
> > > **I worry about the efficiency of the method as for the checks things need to change and model needs to be trained multiple times. This could have a lot of overhead and it needs further analysis.**
> >
> > Thank you for bringing this to our attention. We illustrate below how our approach introduces a negligible overhead compared to a normal training run and that in fact our tests are significantly more computationally efficient than existing attack-based methods.
> >
> > We empirically measured the computation time of the 3 checks we introduced for a CIFAR-10 model trained with DP-SGD on a Tesla T4 GPU. Across 5 runs, we observed that the check for existence of clipping takes $23.12 \pm 1.81$ seconds, the check for per-example clipping takes $17.17 \pm 0.62$ seconds, and the check for noise calibration takes $54.48 \pm 3.07$ seconds. For reference, training with the same setup to convergence requires about 200 epochs, which takes nearly 3 hours. Therefore, the overhead caused by the proposed method is negligible compared to training. It is also significantly lower than existing privacy-attack-based methods which requires training the model to convergence for multiple times (e.g., Tramer et al. (2022) trained 100,000 models on MNIST for privacy auditing, for more details see the last paragraph of Section 2).
> >
> > We also added the following computation analysis to Appendix D and referred to it at the beginning of Section 4:
> >
> > Check for Existence of Clipping. The total number of steps required to complete the test is # training steps $\times $ # Gradient Norm Bounds. We used 1 training steps for 8 Gradient Norm Bounds so the total cost is equal to training for 8 steps. For reference, CIFAR-10 with batch size 100 needs 500 steps per epoch. So the cost of checking is minimal.
> >
> > Check for Per-Example Gradient Clipping. The total number of steps required to complete the test is # training steps $\times$ # mini-batch sizes. We use 1 training step and $100$ different batch sizes. So the cost is similar to $100$ steps of normal training (i.e., $0.2$  epochs of normal training).
> >
> > Check for Noise Calibration. The total number of steps required to complete the test is # models $ \times$ # training steps per model $\times$ # Gradient Norm Bounds, e.g., $2 \times 10 \times 10 = 200$ steps (i.e., $0.4$ epochs of normal training).
> >
> > We summarize the analysis in the table below:
> >
> > Detection for No Clipping | Detection for Mini-batch Clipping | Detection for Incorrect Noise Calibration
> > -------|--------|-------
> > training for 8 steps (~0.02 epoch of normal training)  | training for 100 steps (~0.2 epoch of normal training)  | training for 200 steps (~0.4 epoch of normal training)
> > 23.12 +/- 1.81 seconds   | 7.17 +/- 0.62 seconds  | 54.48 +/- 3.07 seconds
> >
> > > **This is minor: I think the first 3 pages of the paper are very repetitive, without providing much information. This is especially the case for the intro. I think it would be better if the intro actually gave an overview of how you do the check, rather than repeating the stuff that is being checked.**
> >
> > We are happy to revise the writing to remove redundant content.

---

### Official Review · Reviewer_wnYy · 2022-10-31

**Confidence:** 2
**Correctness:** 3
**Technical Novelty And Significance:** 2
**Empirical Novelty And Significance:** 3
**Recommendation:** 5

**Clarity, Quality, Novelty And Reproducibility:**

Clarity: Good

Quality: Okay, the problem studied may not be very important

Novelty: Okay, but not extremely novel

Reproducibility: N.A., I am not an expert on NLP/CV. Thus, I have no idea whether the experiments can be reproduced or not.

**Strength And Weaknesses:**

Strength: The experiment is solid and sufficient

Weakness: I am not persuaded that the studied problem is important

**Summary Of The Paper:**

This paper proposed a method to help the developer to debug the implementation of SD-SGD. Three kinds of bugs can be theoretically guaranteed to be detected: 1. clipping to mini-batch gradient   2. no gradient clipping     3. noise without calibration.

**Summary Of The Review:**

I don't think the studied problem is very important. Since there are already standard/official packages for DP-SGD or other DP optimizers, I don't believe getting DP-SGD incorrectly implemented is a common problem nowadays. Two examples of those packages are TensorFlow Privacy (https://github.com/tensorflow/privacy) and Opacus for PyTorch (https://opacus.ai/).

---

> ### Author Response · Authors · 2022-11-15
> **Response**
>
> We thank the reviewer for their comments.
>
>
> > **Reproducibility: N.A., I am not an expert on NLP/CV. Thus, I have no idea whether the experiments can be reproduced or not.**
>
> The code required to reproduce our experiments is available at https://anonymous.4open.science/r/Finding-Private-Bugs-4087
>
>
> > **I don't think the studied problem is very important. Since there are already standard/official packages for DP-SGD or other DP optimizers, I don't believe getting DP-SGD incorrectly implemented is a common problem nowadays. Two examples of those packages are TensorFlow Privacy (https://github.com/tensorflow/privacy) and Opacus for PyTorch (https://opacus.ai/).**
>
> In addition to being restricted to specific libraries such as PyTorch and TensorFlow, we would like to point out that Opacus and Tensorflow Privacy are often difficult to integrate with other ML codebases - which forces developers to reimplement DP-SGD. For example, the authors of Large Language Models Can Be Strong Differentially Private Learners rewrite Opacus so that it can be integrated with Hugging face’s transformer library, see https://github.com/lxuechen/private-transformers.
>
> Other limitations of these libraries include:
>
> 1. Opacus only supports certain modules and architectures.
>      * Opacus does not support GANs https://github.com/pytorch/opacus/issues/418 and https://github.com/pytorch/opacus/issues/523
>      * Opacus does not support layers typically found in language models and transformers https://github.com/pytorch/opacus/issues/100
>      * Opacus does not support quantized models https://github.com/pytorch/opacus/issues/383
> 2. These packages are not always compatible with other packages, for example CSPRNG cannot be used in Opacus, see https://github.com/pytorch/opacus/issues/504
> 3. Opacus does not support advanced technique such as DP-FTRL https://github.com/pytorch/opacus/issues/393
>
> This is why we believe that it is important to provide simple tests for developers to check for common pitfalls in their DP-SGD implementations. Note that our tests can also be used by developers of DP-SGD packages (for example in a new library that might appear in the future) to verify the correctness of their packages before public release.

---

### Author Response · Authors · 2022-11-17
**Thank you! We (authors) are available for discussion**

Dear reviewers,

Thanks for your time serving as the reviewers for our paper. We really appreciate your comments and suggestions. We are happy to answer any further questions you might have before the response period ends on Nov 18.

Thanks,
Authors

---

### Decision · Program_Chairs · 2023-01-20

**Decision:**

Reject

**Justification For Why Not Higher Score:**

Seems below the borderline really.

**Justification For Why Not Lower Score:**

NA

**Metareview: Summary, Strengths And Weaknesses:**

The paper proposes a methodology for developers to check for mistakes when implemented differentially private ML methods. The reviewers agree that this is an important question, perhaps more to practitioners, but there is not enough interest as the theory is minimal (and at times hard to read) for this to be a paper in top ML conference.